# Latin American and Caribbean Ecotheology: A Kaleidoscope

**Afonso Tadeu Murad *** and **Sinivaldo Silva Tavares ***

Department of Theology, Jesuit Faculty of Philosophy and Theology (FAJE), Belo Horizonte 31720-300, MG, Brazil
* Correspondence: amurad@marista.edu.br (A.T.M.); freisinivaldo@gmail.com (S.S.T.)

**Abstract:** The objective of this article is to present Latin American and Caribbean Ecotheology under the evocative image of a kaleidoscope, in which fragments are combined into a kind of mosaic, suggesting different visualizations. Firstly, a discussion is presented on the relationship between Ecotheology and Liberation Theology, as well as the assumptions of an Ecotheological praxis embodied in the spatio-temporal reality of the continent. The article will emphasize that a kaleidoscopic understanding of ecology—understood as science, paradigm and ethos—reimagines the function and tasks of Latin American and Caribbean Ecotheology. This, in dialogue with environmental sciences and operating a virtuous circularity between academic production and pastoral action, is constituted as theory, practice and spirituality. We emphasize the common elements of Ecotheology and will not dwell on different trends and emphases. We will identify some limits and challenges in this process, without intending to provide an exhaustive analysis. Our aim is to establish reciprocal learning dialogues with Ecotheologies from other latitudes and cultures, and thus effectively contribute to the culture of caring for life on the planet.

**Keywords:** ecotheology; liberation theology; Latin America; kaleidoscope

## 1. Introduction

Ecotheology was born as a Christian response to growing ecological awareness in different parts of the world. It constitutes a fruitful dialogue between the Christian faith and the various facets of ecology. According to Ernest Conradie (2019), Ecotheology consists of a contribution of the Christian faith in addressing the ecological crisis, as well as a proposal for renewal and reform of theology and Christian Churches using an ecological vision.

The identity and mission of Ecotheology are not limited to the action of Christianity in addressing socio-environmental problems, whether emergency or structure-related. Ecotheology encompasses an eco-humanist, critical and proactive Christian perspective. It believes that we receive the Earth from God as a gift and a task. Hence, the emergence of attitudes of gratitude, wonder and praise for the beauty of the planet and the universe, which we call "creation", arises in Christians.[1] We thus recognize ourselves as brothers and sisters of other creatures. We feel called to care for our Common Home in response to the call of the Triune God: creator, redeemer and sanctifier. The interdependence of all beings that inhabit our Common Home is theologically rooted in the theodiversity and mutual collaboration of the Trinitarian community.

Ecotheology consists of a theoretical–practical union that can only be understood in interrelation. A geometric image is like a polyhedron, three-dimensional, with faces, edges and vertices. Or, if we use a dynamic analogy, it is compared to a kaleidoscope, "an optical instrument that serves to create symmetrical visual effects with the help of a set of mirrors and colored glasses". When moving, the reflection of the glass in the mirrors creates beautiful image configurations, with different shapes and colors (https://www.significados.com.br/caleidoscopio/, accessed on 29 October 2023). In addition to its recreational value, the kaleidoscope also serves to provide drawing patterns for artists. The fragments combine to form living and stimulating mosaics. From the point of view of art, "a kaleidoscope allows us to visualize different possibilities, dimensions,

prisms, feelings, emotions, perceptions". It helps to "perceive human beings in their diverse possibilities, formations, intentions, ways of seeing, reading and understanding the context that surrounds them" (da Silva et al. 2015, p. 7). The kaleidoscope of Ecotheology is only realized and allows for multiple configurations when it is configured as a discourse of faith, engaged in dialogue with multiple facets of ecology.

We will reflect on the kaleidoscope of Latin American and Caribbean Ecotheology through the following steps: (1) the relationship of Latin American and Caribbean Ecotheology with Liberation Theology; (2) assumptions for an Ecotheological praxis connected with our time and the reality of the continent (signs of the "spaces-times", interculturation, evangelizing interculturally); (3) the kaleidoscope of Ecotheology in relation to the main facets of ecology (science, paradigm, personal and socio-environmental transformation). This methodological path aims to incorporate some categories that seem more appropriate in characterizing the double movement of Ecotheology on our continent as theory, practice and spirituality. We chose to privilege the common elements of Ecotheology, and we will not dwell on different tendencies and emphases. With Ernst Conradie, we emphasize more the path taken and the trails to follow (Conradie 2019, p. 8). Assuming the existence of a process of growth and continuous "route correction" and seeking to be faithful to its objectives, we will outline some limits and challenges of Ecotheology, without intending to carry out an exhaustive analysis. A complex topic like this, with so many components and configurations, requires synthetic statements to identify the central points and advance the reflection. In some cases, we will use the case of Brazil as an illustration because we believe that a local reality can free us from theoretical abstraction and demonstrate more clearly the challenges and possibilities of Ecotheology.

We hope to make an effective contribution to understanding the path of Ecotheology and encourage sustainable practices, with a view to new models of society.

## 2. Latin American Ecotheology, Daughter of Liberation Theology

The Latin American and Caribbean Liberation Theology (LT) has come a long way since its origins. The work "Liberation Theology" by the Peruvian Gustavo Gutierrez inaugurated and supported something that was being created in ecclesiastical practices in the previous decade (Gutierrez 1987). Leonardo Boff became one of the best-known Liberation Theologians, publishing books such as "Theology of Captivity and Liberation" (Boff 1977b) "How to Do Liberation Theology" (Boff 1986), "Ecclesiogenesis: The Base Communities Reinvent the Church" (Boff 1977a), and "Church, Charism and Power" Boff (1981). João Batista Libanio, with many written works, was notable for his book "Theology of Liberation: Didactic Guide for a Study" (Libanio 1987). Following a different approach from his colleagues, Uruguayan Juan Luis Segundo published books and articles on Christology and Fundamental Theology from the liberation perspective. Among them, "Liberation of Theology" (Segundo 1975) and "Liberation Theology: A Warning to the Church (Segundo 1987)[2]. Among the women who gained prominence in this theological–pastoral movement, articulating it with feminist theology, the Chilean collective Conspirando (starting in 1992) and Ivone Gebara (1997) are worth mentioning.

The celebration of 50 years of Liberation Theology (LT) caused wide-ranging discussions in progressive ecclesiastical circles. We highlight a two-volume work that brings together the reflections of theologians and pastoralists, presenting a broad picture of this theological current, now understood in a plural form ("Liberation Theologies or Liberating Theologies"), and well characterized in the subtitle: "Memory, Review, Perspectives and Challenges" (Guimarães et al. 2022). Below, we highlight some of LT's convictions and striking features.

- Theology is understood as a "Second Act". First comes Christian praxis, understood as a dialectical relationship between communibty practices and methodological and conceptual elaboration. Theology reflects, clarifies and broadens the horizon of understanding for a lived and celebrated faith, ecclesiastical experience and social practices, aiming to overcome poverty and build a just and supportive society.

- Theology and pastoral care take as their "raw material" the "signs of the times", the manifestation and interpellation of God in human society. LT does not simply aim to understand social reality in the light of faith, but rather to transform it.
- LT (Liberation Theology) has a close relationship with the "Church of the Poor", understood as "communities of communities" and "the people of God" and not a clerical structure. The Church of the Poor encourages the formation, action and protagonism of impoverished lay people in Churches and in civil society. The interrelationship between theology (theoretical momentum) and the Church of the Poor (a communal and participatory way of living the faith) is so strong that people and groups engaged in transformative social practices are generically called "members of liberation theology".
- Biblically, LT finds its foundations in the experience of liberation from Egypt, in prophecy, in the liberating figure of Jesus of Nazareth and in the coming of the Kingdom of God. This liberating vision later extends to other books of the Bible. The "Biblical Circles", held to this day in popular communities, constitute a community space for collective appropriation of the word, for thinking about faith and not simply repeating its formulations, for valuing popular wisdom and developing a spirituality centered on Holy Scripture. It also opens space for ecumenical practices, as the so-called "popular reading of the Bible" constitutes a common ground for Churches.
- LT and the Church of the Poor brought with them the attitude of listening to the poor, valuing their wisdom and empathetically understanding their culture and religiosity. Initially there was talk of "inculturation". The Latin American continent is marked by enormous ethnic and cultural diversity, comprising originary peoples, peoples of African descent, the descendants of colonizers and groups of migrants from Europe and Asia. The cultural issue is increasingly becoming a burning subject for theologies of pastoral care and liberation. Yet, the priority is the poor.
- The LT highlights the close relationship between salvation in Christ and historical liberations. Such a vision was affirmed at the Conference of Latin American Catholic bishops in Medellín (CELAM 1968) and ratified in the papal encyclical Evangelii Nuntiandi, on the proclamation of the Gospel, by Pope Paul (1975). From then on, a theological–pastoral process was promoted that highlights the essential social dimension of faith, as opposed to subjective and individualistic views of religious experience.
- Theologians and pastoralists have exercised an essential educational function since the beginning of Liberation Theology and the Church of the Poor, understood in light of Paulo Freire's "Pedagogy of the Oppressed" (Freire 1968), or to use an expression of a sociopolitical nature by Antônio Gramsci, they acted as "organic intellectuals." They accompanied Base Ecclesial Communities (CEB), advised their community assemblies, parishes, dioceses and regions, supported and subsidized social ministries. This task of listening, interpreting and returning reflection to the bases, in dialogue-oriented processes, led to the elaboration of a creative Christian theology, in accessible language, and rooted in the social existence of the poor. Preferentially using the method: "See—Judge—Act", inherited from "Catholic Action", Liberation Theology was not created in university institutions, but in coexistence with popular Christian leaders. Associated with this movement were laymen and laywomen, priests, male and female religious and several bishops.
- Subsequently, LT extended itself to the production of theological works of a pastoral and academic nature. Several books were published in different countries in Latin America. A bold editorial project was then created, entitled "Theology and Liberation" which would comprise 50 books, covering various sectors of theology. Unfortunately, this collective production was not completed because it was restricted by the institutional offices of the Vatican. Only 17 books were published.

- As has already been discussed in several analyses, LT developed distinctively in different countries and regions of our continent. In Brazil, for example, it achieved hegemony for a certain time, even though it was not in the majority. In other regions, it achieved a well-defined space for operating. LT suffered enormous persecution, whether from the economic and political elites of the countries, or from the centralized authorities of Roman Catholicism.

- The Church of the Poor has in its history a large "cloud of witnesses" (Hb 12:1), who shed their blood in defense of the poor and social justice. The martyrdom of popular leaders, priests, nuns and bishops testify that Liberation Theology is not an ideological product of intellectuals, but rather the expression of a living, generous and pulsating community, which lives its faith, exercises charity/solidarity, commits support to causes and nourishes the hope of the poor.

- On the Protestant side, LT flourished in some well-defined Churches, in minority form, especially in the so-called "Migration Protestantism." Among its theologians are José Miguez- Bonino, Julio de Santa Ana and Elza Tamez. In Brazil, Liberation Theology groups stood out among Lutherans (from the Church coming directly from German immigrants), Methodists, a part of the Presbyterian Church and a segment of Anglicans. Among Baptists and Evangelicals, the "Theology of Integral Mission" was established, echoing and expanding the conclusions of the interdenominational Assembly on Integral Mission in Lausanne in 1968 (Fernandes 2019). The main author associated with Integral Mission Theology is the Argentine René Padilla and his work "Integral Mission: Essays on the Kingdom of God and the Church" (Padilla 2014).

- After going through a crisis that shook and brought LT to precious discoveries and new positions (Libanio 2009, 2013), LT assumed a plural identity. It welcomed the identity issues led by various social movements, especially ethnic-racial issues (i.e., black and indigenous), gender issues (i.e., women and LGBT) and cultural issues (i.e., Amazonian and Andean, rural and urban). LT allowed itself to be influenced by the following perspectives: decolonialism, interculturalism, ecological and public theologies. As a result, its horizon of action and its characteristics expanded, as it now speaks of "Liberation Theologies" or "Liberating Theologies" in the web of life (Murad and Tavares 2022, pp. 257–75). It is necessary to consider the different faces of the poor and the complexity of current conflicts, which go beyond the socio-political hermeneutic originating from LT.

After laying out this summarized and incomplete picture, let us now explore the relationship between Latin American Ecotheology and Liberation Theology established at the beginning of the millennium.

The Ecotheology of our continent is similar to a daughter who learns from her mother, internalizes the values received in her own way and follows her own path, yet maintaining her original ties. According to Gutierrez, Liberation Theology includes the classical tasks of wisdom and rational knowledge, offering a critical reflection of historical praxis in light of the Word of God (Gutierrez 1987, pp. 16–18). Ecotheology continues to cultivate this relationship between rational knowledge and wisdom, this time dialoguing with the "ecosophy" and "ecophilosophy" of originary peoples (Estermann 2013) and the ecological paradigm.

- LT protests against the exploitation of the poor. It engages in the struggles of love that liberates in the construction of a new, just and fraternal society (Gutierrez 1987, p. 27). Ecotheology is also committed to these struggles, understanding that the poor and the Earth are wounded and must be healed. According to Leonardo Boff, it is about hearing the cry of the Earth and the cry of the poor as one and the same cry. Such a vision was taken up by Pope Francis in the Encyclical Laudato Si, On Care for Our Common Home. "We are faced not with two separate crises, one environmental and the other social, but rather with one complex crisis which is both social and environmental. Strategies for a solution demand an integrated approach to combating



poverty, restoring dignity to the excluded, and at the same time protecting nature." (Francisco 2015, #139).

- Ecotheology is not content with being merely a theology of praxis which aims at social praxis. It rescues the contemplative character of the science of faith. It does not just act to transform. It also remains silent and reveres the mystery of God in human history and in the cyclical and evolutionary processes of the planet. It combines effectiveness with gratuitousness, effort with enjoyment. It learns from nature to wait for favorable weather, to respect cycles and rhythms. It combines indignation and wonder.
- LT emphasized the community and social dimension of faith. It showed how concupiscence, sin and grace have structural dimensions. It uncovered the political and social implications of following Jesus. It also called on its members to confront the structures that generate and maintain poverty and exclusion. Latin American Ecotheology welcomes and incorporates this project and takes it one step further. It emphasizes its planetary dimension and the interdependence of humans with the Earth's "community of life". This option has consequences for spirituality, the way of formulating and expressing thoughts, and transformative action in the world.
- While Liberation Theology highlights large-scale social and political actions, Ecotheology shows that these are combined with everyday individual attitudes and local collective actions. The various levels of transformative action (individual, community, institutional, political-social) are simultaneous in the process of ecological conversion, as will be explored later.
- Liberation Theology used, in the first moment of elaboration, socio-analytical methods (Libanio 1987), especially critical sociology. Ecotheology also establishes bridges with other forms of knowledge, especially environmental sciences, ecophilosophy, the vision of indigenous peoples and even quantum physics (Boff and Hathaway 2012). Interlocution with environmental sciences enables Ecotheology to format a Christian ecological ethics that is viable, dialogical and scientifically based (Deane-Drummond 2017, pp. 3–9). In line with elements of current thinking, such as 'Complexity Theory", Ecotheological thinking opens unusual windows for knowledge and practice.
- Furthermore, it can be said that Latin American Ecotheology is a different expression and a new offspring of Liberation Theology, at the beginning of this century.

Our next step consists of addressing three important assumptions of Latin American and Caribbean (eco)theology that were put forward by Liberation Theology, in its theoretical and practical aspects, and developed by decolonial theologies.

## 3. Assumptions: "Signs of the Spaces-Times", Interculturation and Evangelization
### 3.1. Discerning the "Signs of the Spaces-Times"

This article proposes "Spaces-Times" as a challenge to theology and the mission of evangelization. Therefore, we will justify the pertinence and relevance of the syntagma "Signs of Spaces-Times". It is obviously not a question of belittling the theological–pastoral fruitfulness of "Signs of the Times". Our objective, therefore, is to suggest a paradigmatic expansion expressed in the syntagma "Signs of the Spaces-Times". Since this is, as it seems to us, a paradigmatic development, we prefer to conceive the relationship between one syntagma and another as a "dialectical assumption" and not as an "overcoming". In this sense, between "Signs of the Spaces-Times" and "Signs of the Times", there is, in our opinion, a process of incorporation, and not of suppression.

### 3.1.1. Would "Spaces" Be Mere Settings for Historical Plots?

Colonial modernity has been imposing itself through two typically colonialist expedients: violent exploitation of the planet's goods and services and the invention of subjects-individuals separated from the Earth. And their relationship with that colonial modernity has been based on exteriority, superiority and instrumentality. The modern colonial invention of subjectivity as cogito has also caused a series of fractures in the natural, social and existential fabric. Separated from Mother Earth, one no longer considers

themselves "sons and daughters of the Earth". Reduced to individuals, we feel separate, opposed and even opposed to ourselves and the other beings that make up the social and cosmic fabric. Finally, one is violently crossed by that existential split that separates us into two things (res): one extensive and the other thinking. One has been, in fact, forced to suffer from a kind of existential, social and cosmic schizophrenia (Tavares 2022, pp. 51–70).

Cartesian thinking served as a glove to the interests of the emerging society concerned with extracting goods from territories and environments, seen as mere resources, transforming them, through the enslavement of human bodies, into commodities available for business of incipient colonial mercantile capitalism. It is worth highlighting that it was not just a question of considering nature as something objective and merely extensive. Human beings of other 'races', considered 'sub-human' or 'non-human', were also treated as simple bodies to be enslaved and subjected to strenuous work.[3] When characterized, above all, by the enslavement of humans of other races and the violent extraction of goods and services from colonized territories—not just by emancipation and autonomy of reason—it was, in fact, colonial modernity that incapacitated us for experiences of "belonging", "respect" and "care" for people, territories and other environmental goods and services.

Colonial modernity has, in this sense, operated an undue reduction of the complexity of reality into two well-defined sections: (1) the objective and inert; (2) the subjective, conscious, free and endowed with moral sense. The section considered "inert" was reduced to a mere setting of the historical plot whose only protagonist would be the human being. This division of roles between both sections is at the basis of the "scientific vision of the world", responsible for denying the historicity and narrativity intrinsic to the world, making the human experience of "being with the world" unfeasible. The setting is considered "exterior" and, therefore, "outside" the plot. This would be the presupposition of those who insist on the notion of "environment" as an external space in which the human being is situated and acts—an autonomous subject detached from his inert setting. This artificial operation would be characterized, for all intents and purposes, as a case of invention, given that on the one hand, the soul is removed from a section of the world, declared objective and inert and, once deprived of any and all activity, reduced to mere scenery. On the other hand, the soul of the opposing section is inflated, declared subjective and considered superior, since it is endowed with admirable capacities for action: freedom, conscience, reflection, moral sense, etc. (Latour 2020b, 116, 142ss).

3.1.2. Space Is the Child of Time!

We are, for all intents and purposes, in the midst of a new epochal configuration. These days, affected by a kind of revolution, what was considered mere "background" has taken up the foreground of our plots. The reason for this radical turnaround seems to be what Bruno Latour unapologetically states: "Organisms make their environment, they do not adapt to it" (Latour 2020a, pp. 162–67).

If, in fact, "space is not a picture, not even a context: space is the child of time, then, strictly speaking, we should get used to talking no longer about history alone, but, rather, about geo-history. Furthermore, if "the space in which we live is exactly the same as the one in which we breathe together", then it will be necessary to admit that space "extends as far as we do; we last as long as those who make us breathe" (Latour 2020a, p. 174). In an interview given to Father Spadaro and published by the Italian daily newspaper La Repubblica, Latour uses terms such as: "construction", "production", "invention" and "fabric" to refer to the relationship between the living organism, Earth, and the living beings that inhabit it:

> When it is explained that the living are those who created the conditions in which they find themselves, it causes a change. The Earth is not living in the New Age sense or in the simplistic sense of a single organism, but is built, produced, invented, woven by living beings. It is not a simple frame within which they move. When I look at the sky above me, its atmosphere, its composition, the distribution of gases, all of this is the result of the action of living beings. A passage opens in which spiritual realities are rich in meaning

*for our earthly condition. The materialism of previous centuries—confirmed as painfully evident—is in fact, very little earthly* (Latour 2022).

In this regard, there is a structural interdependence between all expressions of life on the planet: humans, other living and abiotic beings and entities that populate the cosmos. In this sense, we would like to highlight the lucidity of the "Letter of Earth?" which, by excogitating the neologism "community of life", proposes relationships of belonging, interaction and care for all expressions of life that inhabit planet Earth. This implies privileging relationships and movement, rather than well-defined contours, in a continuous process of composition between organisms, species and collectives.

### *3.2. The Pertinence and Relevance of Doing Theology Interculturally*

The need and urgency for an intercultural transformation of theology has been indicated, expressed in the purpose of doing theology interculturally (Irarrázaval 2002; Herrera Rodríguez 2019).

### 3.2.1. The Choice of the Term Interculturation

Instead of "inculturation"—an older and therefore more frequent term—we chose "interculturation " (Suess 2015). This article does so for two reasons: (1) the impossibility of separating "gospel" from "culture", because, just as every human expression is always linked to a specific culture, the gospel is also born inculturated; (2) the semantic fruitfulness of the prefix inter with regard to what is proposed here as evangelization. Unlike "interculturality"—an abstract noun—"interculturation" takes us to an "action" that expresses the dynamic tension inherent in all dialogue between cultures. There is, in fact, only encounter and dialogue when those minimum conditions are met for relationships of symmetry and reciprocity. These conditions are essential for reciprocal listening and mutual learning.

That said, how can one understand "culture" in the context of "interculturation"? Culture can be conceived as that which is situated beyond multiculturalism and under transculturalism. It goes beyond multiculturalism, since "interculturation" aims at more than just establishing symmetrical relationships between various cultures and managing the relationships that exist between them. However, it is under transculturalism, because it does not share the suspicion that the interaction between various cultures spontaneously and necessarily leading to the dissolution of the properties and peculiarities of each culture. This stance proves to be essential so that the encounter and dialogue between cultures does not degenerate into cultural hybridism and even less into an insipid mixture (García Canclini 1989). By simultaneously placing itself beyond multiculturalism and below transculturalism, the process of "interculturation", therefore, wants to avoid double risks: the creation of ecstatic and monolithic cultures, enclosed within themselves; the defense of globalized cultures with peculiarities and diluted properties.

We also conceive of interculturation in its intrinsic relationship with what one could call "intraculturation ", that is, an action that provides and encourages criticism and reciprocal corrections between cultures in dialogue. In this case, the challenge would not only be to recreate the map of the borders between cultures, affirming the differences, but to establish processes of reconstruction of one's own house, through the recovery of the singularities of each culture, one from the other.

### 3.2.2. Interculturation as "Locus Theologicus"

Interculturation, therefore, would constitute the formal object of theology, affecting its specific method and, therefore, the specificity of theology as a science. The formal object is what makes theology appear as a unique science and, therefore, different from others. It is, in short, what distinguishes it from other sciences. And, as such, interculturation becomes, for all intents and purposes, a "locus theologicus", understood as a starting point and, at the same time, as a perspective of theology itself.

Understood in this way, theology practiced interculturally is not limited to simply affirming or confirming the plurality of cultures. It emphasizes that her theological locus is constituted by the plurality of cultures in dialogue. In other words, it is about symmetrical relationships and horizons between two or more cultures, in order to enrich one another. This relationship is precisely its "inter" quality. In short, it is a theology elaborated from this perspective and from the context of encounter and dialogue between cultures.

Yet, how does one understand the statement of "interculturation as a locus theologicus in its relationship with the classic loci theologici, namely: Scripture, tradition, the magisterium, theological sentences, which are understood as the "sources" of theology? The "theological place" here, should not be conceived in the sense of the classic loci theologici of Melchior Canono, but in the sense of that "place" from and under which the classic theological places of theology are potentialized to the maximum and, therefore, can offer the best of themselves. "Interculturation", therefore, would not only be a place of understanding the data transmitted by ecclesiastical Tradition through the sources of theological knowledge, but rather, it becomes the place in which and from which this data offers the best of itself.

It is in this sense that "interculturation" emerges as a method of transforming theology in its very process of constitution. "Interculturation" is assumed, therefore, as a privileged place and method for rethinking all theological contents and disciplines, including the theology of culture. The assumption of "interculturation" as a locus theologicus causes mutual fertilization in the process of constitution of theological discourse.

The proposals of theology made interculturally in today's theological scenario are found in the exact inversion of the criticisms made by theology itself against theologies that emerged in the context of the last century, all of them under the umbrella of the modern paradigm of history. The criticisms are gathered around three axes: (1) the affirmation of a single history; (2) the imposed and diffused idea of contemporaneity as an expression of the asymmetry between the various temporalities and different times; (3) the assumption of *a* particular history as universal history. Based on this critical dissent, intercultural theology formulates its proposal in three steps or dimensions: (1) respect for different temporalities and the complexity of time; (2) the symmetry between different temporalities and between different "times"; (3) the universal is not separated from the local, nor is it opposed to the particular.

### 3.3. Evangelizing Interculturally

Based on what has been presented so far, we will continue deepening the understanding of evangelization and its systematic discourse, which is theology itself, as a continuous exercise of interculturation, analyzing two of its constitutive attitudes: (1) "dwelling in the boundaries"; (2) "initiating processes"

### 3.3.1. "Dwelling in the Boundaries"

"Dwelling in the boundaries" could, erroneously, give the idea of a static position, as if it were possible to establish "boundaries" in advance, as the delimitation of geographic spaces. On the contrary, "dwelling in the boundaries" precisely wants to express the permanent tension caused by the articulation between three simultaneous movements: exodus, itinerancy and hospitality. This creative tension would be expressed in the prefix inter—inter + culturation—which, in this case, takes us to an intriguing semantic field.

From the literal meaning of border—the delimitation of a geographic perimeter—we are referred to figurative meanings: contours of imaginary spaces. The assumption is that there are environments separated—or connected, depending on the perspective one takes—by a border line. The same line that, in principle, would separate can also distinguish with a view to reciprocity and enhancing relationships.

Understood in this way, "border" would eventually provide an "epistemic subversion" expressed in the following pair of terms: to separate and join; to divide and unify; to delimit and to transgress; to singularize and to differentiate; to confront and to find; to marginalize and to rescue (Raschietti 2022). Ultimately, the demarcation of an entrenched situation can become a condition for the possibility of communication and relationships. The border lines reveal themselves to be porous and, therefore, facilitators of free transit, enabling exchanges and influences.

In order to provoke our imagination, Mozambican writer Mia Couto uses the metaphor of the border, expressed in the prefix *inter*. The excerpt we quote below is taken from his work: "*Rethinking Thought, Redesigning Boundaries*".

> *Our thought, like every living entity, is born to be dressed in boundaries. This invention is a kind of architectural vice: there is no infinity without a horizon line. From the smallest cell to the largest organisms, the design of every creature requires a cap, a separating cover. The truth is this: life is hungry for boundaries. That is how it is and there should be nothing to regret. Because these boundaries of nature are not just for closing. All organic membranes are living, permeable entities. These are boundaries made to, at the same time, delimit and negotiate. The "inside" and "outside" are exchanged in turns* (Couto 2014).

The author suggests a process of authentic "epistemic decoloniality": "learning to unlearn with a view to relearn". These are experiences that translate into attitudes such as, for instance, stripping away all supremacist arrogance—racial, patriarchal, sexist, epistemic— "learning to unlearn" and, at the same time, "rehearsing the relearned" by inaugurating mentalities, attitudes and practices as alternatives to the usual and hegemonic ones. Specifically, it would be about questioning mentalities, attitudes and practices that resort to naturalization and standardization procedures whose purpose would be to legitimize processes of exclusion. And, simultaneously, triggering processes that provide experiences of proximity, inclusion and belonging.

One example, among many, of this stance of "permanent exodus in order to dwell in the boundaries" would be to unmask processes that betray a kind of "invented objectization", namely: God as doctrinal truth; the missionary as an instrument; and the listener as a mere recipient (Raschietti 2022). In this case, "exteriorities" are invented whose scope is the self-affirmation of the institution and its official representatives as protagonists of the mission: objectified doctrinal truths, instruments or simple channels and recipients or mere receivers. The different is denied, using processes of separation, opposition and inferiorization, in order to legitimize exclusionary and supremacist mentalities, attitudes and practices. For this reason, over the centuries, different nomenclatures were considered with the aim of consolidating exclusions: gentiles, pagans, barbarians, foreigners, Indians, blacks, orientals, non-Christians, materialists, modernists, rationalists, empiricists, atheists, agnostics, etc. . .

In response to this "objectification of faith", we remember that the terms "gospel" and "to evangelize", of Pauline origin, which came to occupy a central position in the literary corpus of the New Testament, remind us of the awareness that the Christian proclamation is characterized as "Good News". And, since it is "Good News", it inevitably refers us to the listeners, recognized as interlocutors, in their respective existential, community, social and geo-historical realities. In fact, news is only good if it corresponds in a beneficial and satisfactory way to the needs and expectations of those who hear it. In this sense, it is essential to recognize potential recipients as interlocutors. Their reception and recognition that it is good news becomes a primary criterion for, in fact, the announcement to be recognized as good news. The importance of this awareness is such that "gospel" has become the name of a specific literary genre within the New Testament corpus.

### 3.3.2. "Initiating Processes"

The "new processes" do not appear in parallel to ongoing historical processes, as a kind of appendix or accessory in a competitive position. We are all aware of the complexity of the relationship between established processes and emerging processes. New processes are slowly emerging among the contradictions of ongoing processes.

One has been, unfortunately, conditioned to conceive mission as a kind of expansion and conquest of new spaces. This has been one of the characteristic features of the mission in the "Christian West". It is no coincidence that, until recently, the privileged nomenclature to refer to the mission, especially in official texts, was "implantatio Ecclesiae". In fact, it was a phenomenon of "implantation" with all the consequences inherent to this process. Perhaps in no other dimension has the complicity between Christian Churches and Western culture occurred in such a spontaneous and consolidated way as in this attitude of expansion and occupation of spaces. And, perhaps, this was above all the reason why the mission revealed itself, most of the time, to be an accomplice of Western culture in its structural aspects, namely: colonialist, capitalist, racist, chauvinist and sexist. And, precisely here, lies the structural motivation for why the mission has taken place.

Today, we are aware of the pressing need to initiate processes that lead us to alternative experiences. It is essential to recognize the so-called "recipients of evangelization" as authentic "interlocutors" and, therefore, be willing to learn from them before intending to teach them something ready-made that is imposed on them from outside their cultures and life experiences. Hence, the importance of being close, assuming attitudes typical of itinerant people and pilgrims who approach their interlocutors asking for accommodation and welcome. It is the experience of "feeling like a guest", approaching them with respect, attentive to learning from and with them. It is an inserting of oneself into their world and culture to perceive there "the seeds of the Word" and the subtle interpellations of the Holy Spirit who anticipates us and already actively lives within their lives and cultures.

This implies recovering the original meaning of "evangelization" as an experience of proximity: "feeling with and feeling amongst". And, from there, beginning that journey proposed by the gospels: from conquest to the experience of becoming a guest and itinerant; from expansion to encounter; from implementation and expansion to the experience of encounter and dialogue and, finally, the continuous availability to subvert mentalities, attitudes and practices.

Here, precisely, the necessary dialogue between knowledge, cultures, worldviews and life experiences and customs is inserted and expressed, as in the example of the expression "Ancestral Future" (Krenak 2022). The intention is obviously not to "suspend" the long and winding historical process of the West, and to take refuge in a hypothetical idyllic environment. It is, in fact, about unmasking the illusory myth of unlimited progress within one history, conceived solely and exclusively in its character of linearity, which operates irreconcilable dualisms between: nature and culture, savage and civilized, retrograde and modern, etc.

The alternative character of ancestral temporalities constitutes a reserve of meaning from which to subject the concepts of modern and contemporary to critical evaluation. There are, however, cultural memories and traditions that do not submit to the linearity of historical time. A similar observation puts us before an inevitable question: would not linear history be too narrow to encompass the complexity and future of human existence and life in general? In this precise context and, therefore, aware of the countless questions that arise from this, we propose not only the dialogue between the different temporalities inscribed in the arc of linear and progressive time, but also the dialogue between different times, namely: historical, cyclical and cosmological times Numerous cultures witness fundamental, and not merely foundational, events that occur beyond the conception of time according to the parameters of progressive linearity (Eliade 2011; Panikkar 1967, 1999). Would these cultures continue to be excluded from encounter and dialogue because they do not fit within the narrow limits of Western historical linearity? We propose that the encounter and dialogue between cultures be a privileged place for gathering people's

memories and also a suitable place for exchange and reciprocal enrichment between the various traditions. As Fornet-Betancourt states: "There is not just a plurality of stories, human historicity is temporally pluralistic" (Fornet-Betancourt 2009, p. 99).

Only in this way will we witness the intrinsically inclusive dimension of Christian preaching: each time particular and, therefore, also universal. In the Christian perspective, in fact, there is no experience of universality disconnected from local and particular processes, since the universal does not occur separately from the local nor is it contrasted with the particular. It is essential that any and all attempts at universalization arise from the particularities of each local reality and the singularities of the people and communities there and return to them at the end, maintaining continuous reference to both throughout the entire process. A conception of universality that is not nourished by the encounter and sharing of particular differences will undoubtedly degenerate into imposition and impersonation (Tavares 2018, p. 695–722).

### 3.4. The Presuppositions of Ecotheology

From these presuppositions, one can infer some of the traits of an Ecotheology in process in our continent. Such characteristics are also present in other parts of the world, especially in the so-called "Theologies of the South" and the decolonizing turn that they produce (Tamayo 2017).

- Discerning the Signs of the Spaces-Times in the light of faith is a basic stance of all liberating theology. Reality is complex and many issues, with their varied causes, interrelations and consequences are not evident, or are captured by reductionist ideologies. Hence, there is a need to select the signs and prioritize which are most significant, allowing oneself to be challenged by them. In the case of Ecotheology, the signs "that cry out" are socio-environmental issues, such as the climate crisis; access to water; the loss of biodiversity; the disposal and exponential growth in solid waste and effluents; a consumerism that negatively impacts the environment and people, environmental (in)justice; the threat to food security, the iniquity of the "market society", the relations of colonialisms that are economic, sociopolitical, epistemological and cultural: race, gender, sex, etc. Discerning the "Signs of the Spaces-Times" expands our worldview and leads to decision-making, which translates into gestures, actions and processes.

- The "Signs of the Spaces-Times" are not only negative. The gaze of faith in the God "who makes all things new" (Rev 21:5) leads us to recognize and make visible: good, fair and sustainable practices, the experiences of cultural resistance of subordinated peoples, successful socio-environmental initiatives in the field and in the city, the emergence of ecological spirituality in different religions and religiosities, citizen-driven movements, the advancement of gender, ethnic-cultural and generational identity issues. In all of them, with all their beauties and ambiguities, one recognizes the presence of the Spirit of God who illuminates and renews creation and humanity, leading them to plenitude. Reading the Signs of the Spaces-Times is a contemporary way of exercising biblical prophecy: listening and being attentive to reality, going beyond appearances, discovering meanings, denouncing the ruptures of the Covenant, raising one's voice against the historical forces that produce the degeneration of the planet and the poor, moving from symptoms to root causes, proclaiming hope...

- The syntagma "Signs of the Spaces-Times" alludes to vital issues for originary (indigenous) peoples, communities and Afro-descendants (quilombolas), riverside dwellers, small landowners and families who cultivate with agroecology, the Landless and the movement for community and sustainable ownership and use of rural areas, and those fighting for housing in urban areas. This notion of territory is understood as the intercession of: geographic area, with its specific biome (climate, altitude, type of soil, plant and animal biodiversity), the populations that inhabit it, the culture they received from their ancestors and actualize in the present. The final document of the Pan-Amazonian Synod cites the term "territory" more than 50 times, assuming its

importance for this biome and its residents (Sínodo Amazônico 2019). Maintaining and occupying the territory is more than an economic issue. It concerns the identity of a people and the way they establish relationships with the community of life around them, making history. Thus, as Latour states, the earth is woven by living beings and the human communities that interact with them.

- For Ecotheology, the link between the Creator and the world is not only the human, but rather the entire interdependent set of biotic and abiotic beings. God is in the world, without being confused with it. In God's relational network with his creatures, there are unilateral relationships which: create, conserve, sustain, consummate. And there are others that are reciprocal and configure a cosmic communion of life between the Trinity and creatures which: inhabit, empathize, participate, accompany, endure, delight and glorify (Moltmann 1987, p. 28).

- Our continent is multicultural and multi-religious. One of the plagues implanted by colonization was to introduce into Amerindian cultures the image that they are "backward", "uncivilized", "incapable of governing their destiny". Liberating processes, contrary to the colonialism of the past and the colonial relations of the present, only happen through interculturation. This also marks Latin American and Caribbean Ecotheology. It is in constant dialogue, in a relationship of reciprocity, between the Western way of thinking and the wisdom of the originary peoples, the quilombolas, the mestizo population, the peasants and the poor. Interculturation also manifests itself in learning from the women's movement, generating significant ecofeminist theology and practices on the continent.

- Ecotheology assumes interculturation as a place and method for rethinking theological contents and disciplines. This option implies a long and arduous work of re-elaborating categories, of articulating concepts with narrations and analogies, of testing attempts, of remaking inadequate syntheses, of seeking dynamic balances. She is called to "dwell in the boundaries", taking risks and envisioning her possibilities. To trigger processes that provide experiences of proximity, inclusion and belonging, Ecotheology becomes an "apprentice theology" of socio-environmental movements and liberating contextual theologies.

- Ecotheology shares the mission of Churches to announce the Good News of Jesus and the coming of the Kingdom of God. In this task, it experiences "feeling with and feeling amongst". This position is valid both for humans from different cultures and for other species that cohabit on Earth. The air, soil, water, sun, moon and stars, plants and animals are "others". Inclusivity is perceived and experienced in different dimensions, as both a gift and task.

We will see below how each vertex of ecology interacts with theology, which is also polyhedral and kaleidoscopic.

## 4. Ecology: A Knowledge of Knowledges

Born from biology, ecology broke the tendency to study each species or the physical and biological environment in isolation and adopting only the analytical method. The Austrian physicist Fritjof Capra defines it, in simple terms. It is the study of how Earth-Home (Casa Terra) works, that is, the relationships that interconnect all the residents of Earth-Home (Casa Terra) (Capra 2003, p. 20). Who are these inhabitants? These are the abiotic beings, which do not have life, but are essential for it to happen (i.e., water, soil, air and energy from the sun), and biotic or living beings, from the immense range of species of microorganisms, plants, animals, and us humans, unique mammals. Therefore, ecology does not only study the environment, but the relationships between the beings that coinhabit the Earth, including humanity. It is the science of interdependence. Ecology aims to answer the questions: "how do the mechanisms that govern our 'Common Home' work?" or "how do beings interact to maintain the balance of life on the planet?" and "what does this teach us"?

"The science of ecology studies all interactions between living beings, including human beings, and their environment (...) It emphasizes the study of structures, networks, balances and cycles, rather than causes and effects (...) The objective of ecology is to understand the functioning of living systems in their entirety and not just to decompose them into their constituent elements" (Callembach 2001, p. 58).

By using the word "residents" or "inhabitants", Capra and other authors express a basic conviction of "deep ecology", conceived by Arne Naess: the different beings that "co-live" on our planet are not mere things. The dignity and intrinsic value of all beings and their interactions is recognized. Naess calls them the "relational image of the human being and nature" and the "biospheric equality, in principle", the first characteristics of "deep ecology" (Naess 2007, pp. 98–100).

As ecology develops as a science, it becomes more complex. Addressing, for example, abiotic elements involves studying the energetic factors of light, the chemical factors of water, soil and air and the mechanical-physical factors of gravity, currents and pressure of water and air. Biotic "inhabitants" are understood in the relationships within the same species, in the interaction between different species, in a certain physical and geographic context, in the respective biomes. Given that everything is interconnected, individuals and species cohabit in their own ecosystems (Wagner 1999). All of this together makes up the biosphere; the planet's community of life. This biosphere is increasingly suffering from human influence, causing serious ecological imbalances, to the point of configuring a specific stage in the history of the planet, a geological era that was called by Paul Crutzen and Eugene Stoermer as "Anthropocene."[4]

In several countries, ecology has become a significant area of the tree of knowledge, linked to "life sciences", strengthening ties with other types of knowledge. Following international standards, in Brazil the tree of knowledge in university institutions comprises the areas of specialization: agricultural sciences, biological sciences, health sciences, exact and earth sciences, engineering, human sciences, applied social sciences, linguistics, literature and arts (https://lattes.cnpq.br/web/dgp/arvore-do-conhecimento, accessed on 29 October 2023). Ecology is part of the specialized "Biological Sciences", which includes: general biology, botany, zoology and physiology. As applied ecology, it has a direct relationship with agricultural sciences and their sub-areas such as phytotechnics, soil sciences, forest engineering and management, nature conservation, zootechnics, fishing resources, food science and technology. But this also depends on political and ethical options: accepting the ecological paradigm, or maintaining a merely anthropocentric perspective, based on increasing productivity without taking sustainability into account.

In the area of health sciences, there is a greater interface between nutrition and collective health. In terms of exact and Earth sciences, astronomy and some sectors of physics and chemistry stand out. These sciences help to understand the origin of the universe and the physical–chemical flows of energy and matter on the planet. In the areas of engineering, there is a huge battle with the ecological vision, given the predominance of technocracy and an unlimited conception of progress. At a minority level there are also partners, as we will see below. The interrelationship between ecology and human sciences has grown substantially, given the dialogue with education (the most significant), philosophy, sociology, anthropology, history, geography, psychology, political science, public policies, theology and the sciences of religion.

Another area where enormous conflicts with the ecological vision are found, as well as some positive interfaces, is applied social sciences, which include economics (the main obstacle), law, administration, architecture and urbanism, urban planning and services, communication, social service, tourism and industrial design. The constructive relationship between ecology and these sciences is shown, for example, in eco-communication, ecotourism, ecodesign of products and services, and in urban planning and policy aimed at the population, especially the poorest. Finally, in the area of "linguistics, letters and arts", there is original cooperation. The ecological perception of the world needs to be expressed

beyond concepts, creatively involving the human being with the body and emotions. This area becomes more sensitive to research and communicates a conception of the human being in relation to nature and part of it, through poetry, narrations, theater and other visual arts.

Higher education institutions offer undergraduate, specializations, doctoral and PhD courses in ecology, environmental science, environmental management and other similar titles. Several specific subjects make up the curriculum according to each country and region, such as hydrology, climatology, soil management and conservation, pollution and waste reduction, eco-efficiency, geoprocessing, environmental impact studies, environmental chemistry and physics, etc. Theoretical–practical studies are also developed in other areas of knowledge[5].

In Brazil, as in other countries in Latin America, various branches of knowledge incorporate disciplines and create research groups, in an interdisciplinary, multidisciplinary and transdisciplinary way[6], influenced by the ecological vision: such as beings, species and physical–chemical and biological factors interact, and how to act to reduce negative environmental impacts and increase positive ones, in order to keep the earth habitable.

On the Latin American continent, the growth in ecology science is recent, expanding mainly in the last 30 years. In addition to disciplines with an ecological bias, there is research and publications on environmental law, environmental economics, geography from an ecological perspective (no longer restricted to relief and climate), history and environment, health and ecology, philosophy, etc. Forums, symposiums, seminars and congresses are held in public and private universities around ecology and other areas of knowledge, human, social and exact sciences. Books on sustainability and integral ecology are published from a multidisciplinary perspective, such as the one coordinated by Folmann (2020–2021)[7]. The links with ecology extend to various branches of knowledge. From literature (Mendes 2020b, pp. 99–113) to quantum physics, from the arts to law (Murad et al. 2021). Articles and books are published in which some emerging issues are associated with ecology, such as combating racism, gender claims, confronting poverty, education, decolonialism and interculturality.

If ecology as a science, in its multiple facets, has expanded in the Academy, amid ideological and ethical conflicts, it seems that its reflection does not impact positively on the popular imagination, nor does it contribute to socio-environmental initiatives in civil society. There is a disconnect between the reflection generated in academic spaces and its real influence on the lives of people and human communities. There is a lack of articulation of knowledge (theoretical and operational) and the re-elaboration of knowledge, in order to positively modify people's lives and their lifestyle. And with this, we effectively move towards a new model of an ecologically friendly and socially just society.

On the other hand, there is field research that departs from urban and rural socio-environmental practices, pointing out their contribution to a sustainable society and expanding the scientific horizon in some sectors of the academy. This reality is evident, for example, in community gardens and associations of recycled material collectors in cities, in the recovery of the wisdom of originary peoples and quilombola communities, in agroecology, family farming and solidarity socioeconomic initiatives[8], in the creation of banks of native or creole seeds and the revaluation of regional foods.

## 5. Environmental Sciences and Ecotheology: More Than a Dialogue

One can briefly list some points of the contributions of ecology, especially as a science, to the Christian faith and vice versa. Let us remember that, like a kaleidoscope, the configurations complete each other and are configured in a creative and dynamic way. Thus, the interdependent components of ecology and Ecotheology involve different views and perspectives, including paradigm and ethics (Murad 2022).

- Ecology helps to overcome naive, pre-scientific and negationist discourses about the planet and the seriousness of human intervention on its community of life (biosphere), sometimes supported by fundamentalist religious conceptions. A mature and lucid Christianity, in a plural and wounded world, questions fatalistic and apocalyptic views, which justify the ecological and civilizational crisis as anticipatory signs of the Parousia and the "End of Times".
- By understanding that, to different degrees, everything on Earth is interconnected (interdependence), and that on Earth, the whole is more than the simple sum of isolated parts and that in nature cooperation is more important than competition, Christians are driven to overcome an individualistic faith and strengthen community practices.
- Applied ecology is an important partner for Churches, so that they can take on environmental management in regions, dioceses, parishes, universities, schools, hospitals and social works. Only in this way can "good intentions" be transformed into an effective contribution to society and a credible testimony of love for the Earth and its inhabitants. This responsibility also applies to Christians who work in different companies and fields of work. Without environmental management, ecological discourse can become inoperative and empty.
- The intersection of ecology with scientific knowledge, of a general, strategic and operational nature, provides important knowledge to identify complex causes, create alternatives and choose the most viable ones to deal with the growing degradation of life throughout its entirety on the planet. Such a contribution is fundamental to an actualized and transformative Christian discernment.
- Christian theology, working with philosophy, points out the limits of any scientific knowledge, even if it aims to be integrative and holistic. We cannot fully explain and control nature, as it is surrounded by an inexhaustible mystery, of which we are also a part. The Christian faith brings to ecology attitudes of enchantment, gratitude and respect for the other beings that inhabit our planet. "As part of the universe, called into being by one Father, all of us are linked by unseen bonds and together form a kind of universal family, a sublime communion which fills us with a sacred, affectionate and humble respect." (Francisco 2015, #89).
- Ecotheology, as a public theology, contributes to ecology and other sciences by asking questions and some answers (always partial and necessary) about the "what for" of scientific knowledge, the place of human beings in the world, the indispensability of ethical questions and the relationship between the history of the cosmos and our species. At the same time, it allows itself to be questioned and to learn from and with the other sciences. As a public theology, Ecotheology participates in unusual dialogues for the good of science, politics and mother Earth.
- It is noted that effective dialogues between Ecotheology and other knowledge related to ecology, highlighted in Section 3, are still restricted. There are some occasional initiatives and interdisciplinary research groups, especially in confessional universities. Yet, Ecotheology effectively is still largely limited to dialogue with philosophy, religious sciences and education. This reality is partly due to the resistance of professors and researchers from various areas to carry out dialogue of a religious nature. Furthermore, for a fruitful dialogue it is necessary that the theologian at least has fluency over the most basic concepts, understanding the epistemological status and the linguistic dynamism of other areas of knowledge. In some cases, specific training is necessary, which initially produces an "internal dialogue" within the theologian. Afterwards, there is dialogue with others. As a public theology in the university space, in Latin America, Ecotheology still needs to make enormous progress.

- Theology reveals the limits, ambiguity and fragility of humans, as beings of light and darkness. A reinterpretation of classic theological categories, in an ecological interpretation, such as "creation", "sin", "grace" and "conversion" bring a different perspective on us and the world. This implies an ecological reform of Christianity, rooted in what Conradie calls "the symbols of Christian tradition", comprising the central message of the gospels and the Trinity, the (re)interpretation of biblical and liturgical texts, and the rescue of the virtues of faith, hope and charity.

- The growing threats to the environment and humanity, especially the poorest, lead us to foresee a real scenario of catastrophe for the planet and its inhabitants. Discouragement and feelings of impotence impacts people negatively. Christianity, without denying the gravity of the current moment, attested by science, announces the eschatological hope of God's victory over the forces of evil and "a new heaven and new earth" (Rev 21:1), which are already emerging. The certainty of the Father's goodness, the renewing power of the Holy Spirit and redemption in Christ drive us to engage in "another possible world".

- Along with other religions and religiosities, the Christian faith makes a unique contribution. Solutions to the complex environmental crisis do not come from a single mode of interpreting and transforming reality. "Respect must also be shown for the various cultural riches of different peoples, their art and poetry, their interior life and spirituality. If we are truly concerned to develop an ecology capable of remedying the damage we have inflicted, no branch of the sciences and no form of wisdom can be left out, and that includes religion and the language particular to it." (Francisco 2015, #63).

- According to Román Guridi, religions collaborate in the dialogue on sustainability from three dimensions: (1) they offer narratives, beliefs, motivations and images that legitimize and encourage an ecologically sustainable life; (2) they have archetypes, symbols, meanings and values around which people group and define themselves. Thus, they collaborate to put into practice new habits and lifestyles, confessionally legitimized and communally supported; (3) religions have an institutional impact, as demonstrated by the publication of Laudato Si and Querida Amazônia, within the Catholic sphere[9]. Therefore, it also assumes a critical character, when reviewing images, beliefs and religious narratives that are contrary to the promotion of a sustainable life (Guridi 2022, pp. 359–60). It seems to us that Christian Ecotheology contributes to three dimensions: promoting a sustainable lifestyle, understood in the light of faith, and subsidizing the production of Church documents.

- Ernst Conradie (2020) identifies four basic tasks of Ecotheology, in relation to the ecological crisis and the search for sustainability. The double critique: the ecological critique of Christianity and the Christian critique of ecological destruction. And the constructive double task: contributing to Christian authenticity and to the multiplicity of discourse on ecological issues in the public sphere. It is theologically crucial to discern the movements of the Spirit in the midst of the Anthropocene. According to Conradie, Ecotheology's path to Christian authenticity includes five prophetic and pastoral steps: reading the 'signs of the times' (the symptoms), exposing the underlying causes of the problem (diagnosis), discerning the movement of the Spirit, telling the story of the work of God and expressing a prophetic vision of how the world could and should be. In short: vision and discernment. This path is similar to the method adopted in Liberation Theology and in the "Church of the Poor" in Latin America, which since the Medellín Conference (1968) has adopted the "See-Judge-Act" method, completed with "review and celebrate".

- There are differences and tensions between ecology-science and theology, as the second is legitimized from a tradition centered on the Bible, read and interpreted in the Churches. Even the terminology is not the same. While ecology understands the Earth as an interdependent set of beings in the biosphere, theology considers it as "creation", alluding to a loving project of God. Ecologists speak of "paradigm change" and "overcoming anthropocentrism", while theology emphasizes "ecological conversion",

without denying scientific terms. To encourage dialogue, Ecotheology must adopt the terminology of its interlocutors and know their concepts and categories, pointing out common points and possible differences. Not to entrench itself, but rather to promote the advancement of knowledge and practices.

### 6. The Ecological Paradigm and Ecotheology: Kadeidoscopic Configurations

The expansion of ecology as a significant theme for humanity (one of the most important "signs of the Spaces-Times" of our day) converges with the growth in planetary consciousness, namely, the perception that the human being is the result of a long process of evolution of the cosmos and part of the Earth. Human history becomes one. Thus, the survival of the human species will move towards a convergent history, beyond the borders and interests of each nation, and implement a "planetary governance", in which efforts and alternatives are combined and intertwined. We are in a great "Noah's Ark": either we will all survive or, together, we will perish.

The blossoming of planetary consciousness calls into question the vision of human beings and the way they act in relation to the Earth's community of life (biosphere). This leads to a turn in the kaleidoscope of ecology, now understood not only as the interdependent science about the Earth and humans, but as a paradigm or macro model for understanding the Earth and humanity.

The contemporary use of the term "paradigm" originates from Thomas Kuhn (originally used in 1962) who fundamentally understands it as an organizing model of scientific knowledge. With this concept, Kuhn shows that science does not evolve linearly. It happens in the form of leaps, when a previous paradigm is surpassed by another, which manages to gather data in an innovative way and thus enable changes. Every paradigm involves a choice by the scientific community, highlighting a certain way of organizing information and underestimating others. Therefore, a paradigm is provisional, although essential.

The word "paradigm" went beyond the scientific scope and also came to mean "model of understanding" in general. This model has different scopes, thus configuring microparadigms, mesoparadigms and macroparadigms. For example, replacing the mechanical wristwatch with a digital one would be a microparadigm change. Overcoming modern anthropocentrism constitutes a macroparadigm, as it implies profound changes in vision.

In this broad sense, the ecological paradigm questions the egoic and unbalanced anthropocentrism which came about through modern science and was consolidated with technoscience. This brought many solutions, but created several problems, to the point of making ecosystems and human society more vulnerable. According to Riechmann, "the current ecological crisis results from mismatches in the interaction between the biosphere and technosphere (...) The linear processes that govern the industrial technosphere clash violently against the cyclical processes that prevail in the biosphere". These absorb more and more matter and energy and excrete waste at an unsustainable rate (Riechmann 2005, p. 114). Thus, "our way of life as a whole—our way of working, producing and consuming—is not durable over time, nor is it generalizable to all inhabitants of the planet" (Riechmann 2005, p. 46).

The ecological paradigm presents different expressions and accents, sometimes in conflict. Biocentrism, ecocentrism and inclusive anthropocentrism stand out. With different emphases, they consider the value of each being for the continuity of life on the planet, as well as recognizing, to different degrees, the uniqueness of the human species.[10] Inclusive anthropocentrism emphasizes the uniqueness of our species, characterized as a being of language that creates meanings, makes history, acts on the environment and carries out civilizing processes. Biocentrism and ecocentrism emphasize the intrinsic value of each being that constitutes the Common Home and their relationship of interdependence and cooperation in ecosystems. Other authors, such as Sinivaldo Tavares, prefer to abandon this distinction and adopt a relational perspective typical of networks or webs, in which the plot is woven not around a single center, but whose threads are woven around several 'nodes'.

The emerging ecological paradigm also presents an original contribution to epistemology. Such a turn positively impacts the experience of faith and theological knowledge. It is knowledge that involves the knowing subject (Moltmann 1987).

Leonardo Boff expanded the horizon of Latin American Liberation Theology. Among his Ecotheology works, we mention here: "The Earth in the Palm of your Hand: A New Vision of the Planet and Humanity" (2016), "Caring for the Earth, Protecting Life" (2010); "The Painful Birth of Mother Earth" (2021). The foundational work is "Ecology: Cry of the Earth, Cry of the Poor", translated into several languages and with consecutive editions published in Brazil. The first edition with this title, from 2004, was revised and expanded in 2015. Leonardo does not elaborate a "theology of creation" nor a "green theology". He considers ecology as a new model of understanding human beings and the cosmos (paradigm). As a visionary Liberation Theologian, he articulates the environmental issue with the social issue, considering them in a unitary and plural way. Due to its paradigm conception, it configures Ecotheology as something more than a dialogue between socio-environmental themes and Christian faith. This can be understood in light of the ecological paradigm and vice versa. It is observed that Boff practices 'Ecotheology', but uses this word little in his works.

According to Boff, "ecology is a knowledge of relationships, interconnections, interdependencies and exchanges of everything with everything at all points and at all times". It does not simply consist of knowledge of objects of knowledge, but of relationships between objects of knowledge—Professor and Researcher. Department of Theology. FAJE (Jesuit Faculty of Philosophy and Theology). 31.720-300 Belo Horizonte MG Brazil.a "knowledge of knowledges" related to each other. Ecology aims to understand the way in which beings depend on each other, in an immense web of interdependence, the homeostatic, balanced and self-regulated system. The uniqueness of ecological knowledge consists of its transversality, that is:

> (. . .) relating lateral (ecological community), forward (future), backward (past) and inward (complexity) all experiences and all forms of understanding as complementary and useful in our knowledge of the universe, our functionality within it and in the cosmic solidarity that unites us all (Boff 2015, p. 17).

Holism results from this stance, as it translates the capture of the organic and open totality of reality and knowledge about this totality. The set of living organisms and their environment co-evolve simultaneously. The whole is more than the sum of its parts (holism), and in each part of the universe the whole is present (holographic principle) (Boff 2015, p. 35).

Boff expands Thomas Kuhn's concept by defining paradigm as "an organized, systematic and coherent way of relating to ourselves and everything else around us. These are models and standards of appreciation, explanation and action on the circumcising reality" (Boff 2015, p. 24).[11] Ecological paradigm emerges from the planetary community, characterized by a unifying vision of the human being and the cosmos, as humans "are a moment in an immense process of universal interaction". Moreover, the Earth is considered to be an extremely dynamic and complex organism, which has identity and autonomy. It is the great mother who nourishes and carries us.

There is a confluence of the evolution of the cosmos with the path of humanity. Historicity does not only concern the human being, but also the universe and nature, through cosmogenesis. The rigid separation between nature and history, world and human being, which has legitimized so many dualisms, is naive. The destiny of humanity is linked to the destiny of the cosmos. We belong to each other. Following Teilhard de Chardin, Boff maintains that in the human species, interiority and complexity have gained self-conscious expression. The human being is a child of the Earth, and the Earth itself an expression of consciousness, freedom and love.

An attitude of wonderment, a new sacredness, a feeling of intimacy and gratitude then emerges. The universe of the living fills us with respect, veneration and dignity. Alongside instrumental reason, symbolic and cordial reason is valued; all our bodily and spiritual senses are used, for we are logos (reason), and affectivity (pathos), desire (Eros), passion, commotion, communication and attention to the voice of nature that speaks in us (daimon). Knowing is not only a way of mastering reality, but also of entering into communion with other beings (Boff 2015, pp. 28–29).

The emerging paradigm requires a new logic, which connects and relates both human and universal realities to each other in a dialectical, reciprocal and perichoretic manner. This perichoretic logic seeks dialogue in all directions and at all times. It is the most inclusive attitude possible due to the fact that it produces the smallest number of victims. Everything interacts with everything, at all points and in all circumstances.

Ecological thinking welcomes advances in the scientific method, but proposes something more. It considers beings, organisms and phenomena in the set of inter-relationships that concretely constitute them. A being can only be known by taking into account the ecosystem and the web of its relationships. Hence, knowing the part in the whole and the whole present in the parts (Boff 2015, p. 45).

The emerging paradigm offers the conditions for a holistic spirituality, as it advances into the spiritual depth of the universe. The rigid separation between "natural sciences" and "spiritual sciences" is broken. From quantum physics, it is stated that the phenomenological side and that of interiority/spirituality are two dimensions of the same world. The spirit belongs to nature and nature appears spiritualized. From there, we move towards a pan-en-theistic conception. In other words: the material world is not Divine, but God is present in it. Beings and the systems that constitute them have a certain level of life and spirit. Everything has interiority (Boff 2015, pp. 49, 51). As J. Moltmann states, God is present in creation, through his life-giving spirit, without being confused with it (Moltmann 1987).

In later works, Boff advances the reflection and brings together the characteristics of what he calls the "eco-cosmological paradigm" (Boff and Hathaway 2012). According to Boff, the Common Good extends to the planetary and cosmic community of life. This connection is due to the fact that everything that exists and lives deserves to exist, live and coexist. The holistic–ecological attitude encourages us to be "increasingly unique and at the same time supportive, complementary and creative". In another work, Leonardo develops the theme of care, understood as "a way of being in the world that founds the relationships that are established with all things, that originating force that continually gives rise to the human being". Essential care comes from listening to the Earth and from passion for the planet and people (Boff [1999] 2002, pp. 92, 101). Caring for people and the Earth "implies having intimacy, feeling them inside, welcoming them, respecting them, giving them peace and rest. Caring is getting in tune, listening to their rhythm and tuning in with them" (Boff [1999] 2002, p. 96).

What is being carried out in pastoral action and theological production, in the sense of allowing ourselves to be touched by the ecological paradigm and contributing to its development? In what sense does the ecological paradigm encourage theology to advance in its reflection?[12] What footprints signal a path traveled and to be followed?

Perhaps our analysis is pessimistic. But there are clear signs that, with exceptions, the evangelizing action of Churches and the teaching–learning of academic theology have not yet made the leap to the ecological paradigm. We have already made some observations in Section 4 and we will complete it with the following observations below.

- For Christians, adopting the emerging paradigm involves an "ecological conversion". This implies understanding the planet's community of life (ecosystems, biomes and biosphere) and human communities in an interdependent and relational way. We are not kings of nature, but rather an eco-dependent species, as we need water, soil productivity, food, breathable air and climate balance to continue to inhabit the Earth. A change of mind, heart and hands, which requires effort, abandonment of unsustainable habits and new learning. One must implement the transition from the conception

that we would own, dominate and plunder the planet, to that of managing, cultivating and caring for the Earth. "This implies a relationship of mutual responsibility between human beings and nature." (Francisco 2015, #2,#67).

- It is therefore necessary to adopt another lifestyle, simpler, less consumerist, in tune with nature and enhancing collaborative human relationships, what Francisco calls "happy sobriety" (Francisco 2015, #222–26). In the same vein, leaders of alternative movements and spiritual leaders propose "voluntary simplicity", "full consciousness" and intercultural and planetary solidarity.

- Ecological conversion simultaneously requires collective, institutional, economic actions (i.e., the way of extracting, producing, transporting, selling, buying and discarding products) and policies at local, regional, national and planetary levels. For Christians, this means considering the ecological commitment as an essential part of their vocation in the world, acting in a communal fashion. In Francisco's words, "self-improvement on the part of individuals will not by itself remedy the extremely complex situation facing our world today... Social problems must be addressed by community networks and not simply by the sum of individual good deeds. '[It] calls for a union of skills and a unity of achievement that can only grow from quite a different attitude'. The ecological conversion needed to bring about lasting change is also a community conversion." (Francisco 2015, #219).

- The Synod for the Amazon discussed that the different dimensions of conversion are intertwined. This change in attitudes and practices is based on following Jesus, now lived with a more comprehensive perspective. The only conversion to the living Gospel, which is Jesus Christ, develops in interconnected dimensions to motivate a "Church which goes forth" to the existential, social and geographic peripheries of the Amazon. New paths of pastoral (chap II), cultural (chap III), ecological (chap IV) and synodal (chap V) conversion are then proposed (Sínodo Amazônico 2019, #19). The conclusions of the Amazon Synod are inspiring for other contexts as well.

- What is the real incidence of Ecotheology in academic or seminary theology courses? It does not seem to us that it is not yet visible and recognizable. As an optional subject (not mandatory) it appears in the curriculum of few Theology Faculties. Sometimes it is restricted to a chapter of Social Morality. These steps, while still timid, must be supported. However, the discipline requires a method consistent with the ecological paradigm. In other words, it associates reason and emotion, promotes a sustainable lifestyle, creates sensory experiences in tune with nature, encourages ecological spirituality and suggests personal and community ecological practices. Only in this way can it contribute to ecological conversion.

- The main challenge is to make the ecological paradigm fertilize various areas of study and theological disciplines. The basic question is: "how do other creatures and the entirety of creation participate in God's salvific project"? Although there is a growing production of theological themes from an ecological perspective, it is not used sufficiently by theology students and is not included in the list of bibliographical references for professors.

- The interdependent perspective of ecology (and Ecotheology) requires work to interrelate the different disciplines and areas of study in theology courses. It is not enough for teachers to know the content worked on by their colleagues. It is necessary to carry out pedagogical experiences of interdisciplinarity, which help to overcome (or at least reduce) fragmentation and promote a "tour" through theology.

- In several countries on our continent, theology is recognized by official bodies, at undergraduate, masters and doctoral levels. Although it is part of an area of study with other humanities courses, interaction with the university is restricted. It is necessary and urgent to learn from other regions of the world, where theology students can build part of their curriculum by attending courses at other faculties. For Ecotheology, this dialogue with environmental sciences would be fundamental in the process of theological studies. There are many possibilities, as we listed in Section 3.1.

- As Boff highlights, the ecological paradigm is not limited to an intellectual vision of the interdependent relationship between humans and the planet. It comprises attitudes of respect, reverence, cooperation and care for Mother Earth and people, especially the most fragile and invisible. Care translates into relationships of welcome, preservation and regeneration. We must question how our ecclesiastical communities cultivate this care and creatively stimulate its practice, according to local realities.

## 7. Ecotheology in Pastoral Care and Transformative Practices

Latin American Liberation Theology elaborated the distinction and articulation of the different levels of theological discourse. Clodovis Boff calls them "popular", "pastoral" and "professional" theology.

*"Liberation theology can be compared to a tree. He who only sees professional theologians in it sees only the branches of the tree. He doesn't even see the trunk, which is a reflection of the shepherds and other pastoral agents, and even less does he see all the roots that are underground and support the entire tree: the trunk and the branches. Although underground and anonymous, this is the vital and concrete reflection, of tens of thousands of Christian communities, who live their faith and think in a liberating way [...]. What unifies these three planes of theological-liberating reflection? The same underlying inspiration: a faith that transforms history, or, in other words, concrete history thought from the leaven of faith. The same sap that runs through the branches of the tree is also that which passes through the trunk and rises from the secret roots of life."* (Boff 1990, pp. 91–92).

This contribution is valid for any theology at the service of evangelization and dialogue with the contemporary world, and especially for Ecotheology. If a certain academicism still survives in theological production in general, it is necessary, however, to emphasize that on our continent there is a movement "from the bottom up", involving leaders who work in different areas. Pastoral Ecotheology effectively illuminates the actions of Christians committed to socio-environmental organizations, whether sponsored or led by Christian Churches, or present in socio-environmental organizations in civil society.

If ecology is the knowledge of interdependence and cooperation, Ecotheology consists of the knowledge of relationships that, in the ecclesiastical context, brings together and articulates the spirituality of concrete communities, pastoral practice and the intellectual elaboration of knowledge. Theological thinking has different levels, depending on the recipients, interlocutors, existential situations, social context and pedagogical purposes. In this regard, João Batista Libanio and Afonso Murad express this theological thinking as follows: "everyday", "popular" and "pastoral" theology (or "theology in pastoral care", or "theological initiation") and academic professional theology (Libanio and Murad 2014, pp. 191–99).

Everyday or popular Ecotheology happens on a daily basis, as a spontaneous expression of the existence and spirituality of people and groups, located in the respective biomes. It manifests itself in popular wisdom, in narrations and songs. When an indigenous community needs to remove a tree to make a canoe or a wooden artifact, they ask for permission and show gratitude. A family that cultivates the soil in an agroecological way knows that insects should not be destroyed with chemical poison, but rather controlled. The community praises God and celebrates at the time of the corn or bean harvest. It recognizes that everything is a gift and that we humans cultivate and care for the garden of creation in this way, even if the community does not express it in those exact words.

What is characteristic of "pastoral ecotheology"? It takes place in assemblies, meetings, training courses, preparation of educational material (i.e., booklets, texts, magazines) especially for community and social pastoral leaders. In accordance with the methodology inherited from "Catholic Action"—"See, Judge, Act, Evaluate and Celebrate"—Ecotheology intervenes after analyzing the reality in question, providing theoretical–practical illumination, in the light of the Bible and contemporary studies. The schemes, texts and presentations of the advisors are presented in a dialogical way. According to the report of

experiences, the theologian modifies the previous content he had prepared, in order to keep it as something significant for his interlocutors. Pedagogically, images, analogies, songs, dances and narrations of experiences are used. The prayerful moments reveal a spirituality that has to do with local cultures and nature. Sometimes celebrations are held outdoors and in the shade of trees. Symbols refer to our communion with God and nature.

For illustrative purposes, let us explore some examples.

According to its official website (https://www.amerindiaenlared.org, accessed on 29 October 2023) Amerindia is a network of Catholics with an ecumenical spirit and open to interreligious dialogue and cooperation with other institutions. Operating as a foundation based in Montevideo, Uruguay, it aims to reaffirm the preferential option for the poor and excluded, inspired by the Gospel, responding to the new challenges posed to our countries by neoliberal globalization. This implies reaffirming the option for new models of community and participatory Church and Liberation Theology as a contribution to the Church in the world.

Amerindia originated in 1978, on the occasion of the preparation of the Third General Conference of the Latin American Episcopate in Puebla, Mexico, by a group of theologian advisors. From 1998 onwards, it became a wider, plural and representative network of bishops, theologians, communicators, educators, social scientists, religious and lay people, committed to the Church, new movements and social actions, reaffirming the preferential option for poor and actualizing the theological–pastoral heritage of Latin America and the Caribbean. With a history woven by many hands, "from God and from below", Amerindia has consolidated itself as a liberating ecclesiastical current, articulated around a new way of being and acting, based on solidarity with the cry of the excluded and the mother land (https://www.amerindiaenlared.org/sobreamerindia/, accessed on 29 October 2023).

Amerindia promoted three "Continental Theology Congresses", aimed equally at theologians and leaders of Base Ecclesial Communities and social–pastoral ministries. It offers subsidies for reflection and organizes publications that update Liberation Theology in this context at the beginning of the century[13]. Amerindia's latest publication is titled "Susuros del Spiritu. Theological density of them liberation processes" (Aquino Júnior et al. 2023). This voluminous work of 747 pages integrates the contributions of ecofeminism, Afro-theology, indigenous peoples, decoloniality and the challenges of the urban context. It includes Ecotheology themes, such as ecological spirituality and migration. It is an example of how professional theology and "other perspectives" articulate well with emerging pastoral themes, further developing Liberation Theology.

At the "Ameríndia" meeting in the Amazon region in June 2023, theologian Sinivaldo Tavares proposed a reflection on the "Signs of the Spaces-Times", interculturation and, after hearing reports from experiences of pastoral consultancy and theology from different countries. That day, participants spent the entire journey on a boat, in order to experience the beauty and immensity of the river and forest.

Another social pastoral organization, linked to the Conference of Bishops of Brazil, carrying out its work in line with Ecotheology, from a socio-environmental perspective, is the Pastoral Land Commission (CPT). The Commissiom was born in June 1975, during the Meeting of Bishops and Prelates of the Brazilian Amazon. It was founded as a response to the serious situation experienced by rural workers, especially in the Amazon, exploited in their work, subjected to conditions similar to slave labor and expelled from the lands they occupied. The commission's action extended to regions of Brazil where farmers in their most diverse categories faced similar problems. In each region, CPT's work took on a different tone according to the challenges of reality (https://www.cptnacional.org.br/sobre-nos/historico, accessed on 29 October 2023). CPT is the only organization in Brazil that, each year, presents a picture of land conflicts in the country, denouncing socio-environmental injustice, violence and the murder of indigenous leaders and field workers. CPT promotes the campaign to combat slave labor, with the participation of several partners, the articulation of sustainable practices in the Cerrado (savannah biome) and the Amazon.

As it delved deeper into the cause of equitable land use, CPT took on an Ecotheological vision. This does not come as a complement or "something coming from the outside", but is already part of their practice, discourse and spirituality. The CPT regional offices promote the Earth and Water Pilgrimage every year. It is a religious event, in which participants walk and pray, sing songs and perform symbolic gestures. They praise God for the gifts of creation, give thanks for the strength of the union, denounce the depredation of water and soil and nourish hope (https://www.cptnacional.org.br/, accessed on 29 October 2023). At the same time as it rescues elements of popular religiosity, it promotes the growth of socio-environmental awareness and encourages commitment to struggles in defense of water, soil and people.

In the Latin American context, we highlight an ecclesiastical organization in which Ecotheology offers special collaboration and, at the same time, learns from them. This is the Panamazonic Ecclesial Network (REPAM). It emerged in September 2014 and established its headquarters in Quito, Ecuador, being transferred to Manaus, Brazil in 2022. It presents itself as "a platform for synodal articulation, sharing experiences and services to respond to the needs of the territory of eight countries and one overseas territory of the Pan-Amazon (Bolivia, Brazil, Colombia, Ecuador, Guyana, French Guiana, Peru, Suriname and Venezuela)". It is a network that cultivates, cares for and strengthens horizontal processes with the people and ecclesiastical organizations of the Amazon. It aims to incarnate itself in the daily lives of people, promoting committed and sensitive meetings, as well as moving other spheres of awareness and advocacy (https://www.repam.net/pt/quem-somos/, accessed on 28 October 2022).

As an Ecclesial network, it assumes the proclamation and construction of the Kingdom of Life based on the Gospel. REPAM makes the Church's activities in the Amazon territory visible and mobilizes the efforts of different organizations, pastoral groups, movements, congregations and other ecclesiastical actors. The territory of its action is the Amazon basin, which covers nine South American countries. This space understood not only as the biome, but also as the place where different people build and carry out their life projects (https://www.repam.net/pt/quem-somos/, accessed on 29 October 2023).

The Amazon, as stated in the Pan-Amazon Synod, is a beautiful, fragile and complex, multi-ethnic, multi-cultural and multi-religious reality (Sínodo Amazônico 2019, #8). It demands an open dialogue in view of sustainability and "Good Living" (Buen Vivir), from various interlocutors: "the indigenous peoples, the river dwellers, peasants and afro-descendants, the other Christian Churches and religious denominations, organizations of civil society, popular social movements, the State, finally all people of good will who try to defend life" (Sínodo Amazônico 2019, #23). Bishops, representatives of indigenous peoples, pastoral agents and religious institutes linked to REPAM played a fundamental role in the Pan-Amazonian Synod, which outlined new paths for the Church in the region, in line with integral ecology. The final text of the Synod requests that the formation of priests have an Amazonian face. It proposes that academic training be included: integral ecology, Ecotheology, theology of creation, Indian theologies, ecological spirituality, the history of the Church in the Amazon, Amazonian cultural anthropology, and so on" (Sínodo Amazônico 2019, #108). Pastoral Ecotheology and ecospirituality are present not only in chapter IV (Sínodo Amazônico 2019, pp. 64–85), but have an influence over the entire document.

Aiming to operate close to the territories, REPAM is divided into national networks, which have different degrees of action and impact in their countries. There are also several thematic groups, such as those concerned with "Socio-Environmental Justice and Good Living". At a regional level, REPAM Brazil promoted three meetings, bringing together leaders who work in the Amazon region. Several topics covered at these meetings were published online, starting in April 2021, in "Revista Ecoteologia."[14] The magazine combines articles of pastoral content with others of academic language. REPAM's focus, as we have already said, is not ecology or Ecotheology. However, these both present themselves as a connecting element for the multiple issues in the Amazon.

*Ecotheology Pastoral: Short Reflections*

- Fortunately, there are several initiatives on the continent in which Ecotheology is present as one of the inspiring lines, learning and responding to new questions, such as the "Missionary Indigenous Council" (CIMI)[15] and "Network Churchs and Mining"[16]. Such initiatives are modern expressions of Liberation Theology and the Church of the Poor. They work with minority groups with strong socio-environmental commitments. Yet, this work does not have much impact on social media. This poses an urgent task, namely: adapting the language, entering the world of modern media, raising awareness, and countering neoconservative groups, expanding the range of sympathizers and supporters.
- The concept of "Integral Ecology" is widely used in Latin American and Caribbean Ecotheology, both at pastoral and academic levels. In theology it was introduced by Leonardo Boff, who added integral ecology to Guattari (1990)'s three ecologies. Subsequently, Pope Francis assumed integral ecology as one of the bases of the Encyclical Laudato Si, choosing it as the transversal axis of the document. Furthermore, by making it a central theme in chapter IV, the Pope proposes it as an articulator of the different facets of ecology: environmental, economic, political, social, cultural and people's daily lives, especially in an urban context. Integral ecology also concerns the "Common Good", a classic theme of the Social Doctrine of the Catholic Church, and intergenerational justice (Francisco 2015, #137–62).
- As happens in other parts of the world, Ecotheology infects and enriches the various sectors of theology, whether in academic production itself or in reflections related to intercultural evangelization. On our continent, production in the area of the Bible stands out. There are also relevant works in systematic theology, such as: theology of creation, theological anthropology, Christology, Trinity, Eschatology, theology of religions, liturgy and sacraments. There are reflections that integrate Ecotheology with liberating contextual theologies, such as Indian theology and Black (or Afro) theology. The main conceptual advance occurs when Ecotheology establishes a close relationship with decolonial and feminist perspectives (ecofeminism) as well.

## 8. Conclusions

This article has shown that Latin American and Caribbean Ecotheology is a daughter of Liberation Theology (LT) on our continent. From the latter, it learns to theologize through practice and promotes processes of greater awareness, aiming to form people and groups that work towards socio-environmental transformations. Ecotheology incorporates ecology within the framework of social issues, such as overcoming poverty and participation in citizen-driven movements and community organizing. The two aspects become one: socio-environmental. The first phase of LT sought to overcome modern colonial anthropocentrism, showing that it was exclusionary, as it did not consider the invisible faces of poor people, indigenous people, people of African descent and women. Ecotheology takes a further step by recognizing the "cry of the Earth" and calling for justice in the relationships between humans and nature. Pastoral Ecotheology developed LT's intuition of cultivating an integrative relationship between rational knowledge and wisdom. It values transformative praxis as well as gratuitousness, praise and celebration. Ecotheology dialogues not only with social sciences, but also with the environmental sciences. It proposes "ecological conversion" as a transformation process that simultaneously affects everyday lifestyles, sustainable productive structures, views of the human being as part of the earth, responsibility for the continuity of life in all its extension. This shows the beauty and multiple possibilities of its kaleidoscope.

We articulated Ecotheology with three pressupositions that we consider fundamental in evangelization on our continent. In the midst of a complex, sometimes confusing and dispersive reality, we illustrated the indispensability of discernment about "spaces-times". The relevance of the interculturation processes involves: the recognition of "others" and, in differences, inaugurating learning and sharing relationships. On a multiethnic, multicul-

tural and multireligious continent, evangelization will always take place interculturally. This plurality is an indispensable condition for it to become "Good News" and, therefore, exempt from the weight of domination. These assumptions directly impact the Ecotheology of our continent, in its contents, transmission and reception processes.

Finally, we considered that the kaleidoscopic understanding of ecology—understood as science, paradigm and ethos—reimagining the function and tasks of Ecotheology as theological knowledge in dialogue with environmental sciences. It is about understanding the human being, in the light of faith, as a brother or sister to other creatures and a promoter of socio-environmental justice. It was emphasized that Latin American and Caribbean Ecotheology operates a virtuous circularity between academic production and pastoral action. The challenge remains to re-elaborate the eco-theological discourse and its spirituality in a language accessible to the vast majority of the population, which moves within the horizon of popular devotion, mixed with religious traditions of various origins.

We hope that Latin American and Caribbean Ecotheology, based on its originality, establishes dialogues of reciprocal learning with the Ecotheologies of other cultures, peoples, languages and nations. This will make this theological movement, with different emphases, more than a "contextual theology". It will become an effective contribution to the culture of care for life in the world.

**Author Contributions:** Conceptualization, A.T.M. and S.S.T.; methodology, A.T.M. and S.S.T.; software, A.T.M. and S.S.T.; validation, A.T.M. and S.S.T.; formal analysis, A.T.M. and S.S.T.; investigation, A.T.M. and S.S.T.; resources, A.T.M. and S.S.T.; writing—original draft preparation, A.T.M. and S.S.T.; writing—review and editing, A.T.M. and S.S.T.; visualization, A.T.M. and S.S.T.; supervision, A.T.M. and S.S.T.; project administration, A.T.M. and S.S.T.; funding acquisition, A.T.M. and S.S.T. All authors have read and agreed to the published version of the manuscript.

**Funding:** This research received no external funding.

**Institutional Review Board Statement:** Not applicable.

**Informed Consent Statement:** Not applicable.

**Data Availability Statement:** Data are contained within the article.

**Acknowledgments:** Special thanks to Michael A. Martínez SJ, for translating and stylistically reviewing this article.

**Conflicts of Interest:** The authors declare no conflict of interest.

## Notes

[1]    "In the Judaeo-Christian tradition, the word "creation" has a broader meaning than "nature", for it has to do with God's loving plan in which every creature has its own value and significance. Nature is usually seen as a system which can be studied, understood and controlled, whereas creation can only be understood as a gift from the outstretched hand of the Father of all, and as a reality illuminated by the love which calls us together into universal communion." (Francisco 2015, #76).

[2]    The book by Juan Luis Segundo had in its original Spanish version the following subtitle: "Response to Cardinal Ratzinger". Segundo's text is a dense and theologically well-constructed text, presenting the deficiencies of the "Instruction on Certain Aspects of Liberation Theology" (1984). The text from the Congregation for the Doctrine of the Faith had a negative impact on the continued development of LT.

[3]    In this regard, we refer the reader to the incisive reflections of the Peruvian sociologist, Aníbal Quijano, regarding the "invention of the idea of race" (Quijano 2000, pp. 201–46; 2007, pp. 93–126).

[4]    See the analysis by N. Wirzba (2023, pp. 34–64), in the section "Facing the Anthropocene". Also see the brief presentation on the topic in (Mendes 2020a, pp. 1–4).

[5]    For example, a work used in Brazil in Engineering and environmental management courses. The book, entitled "Introduction to Environmental Engineering: The Challenge of Sustainable Development" comprises three parts with respective chapters: (I) Fundamentals: enavironmental crisis, laws of conservation of mass and energy, ecosystems, biochemical cycles, population dynamics, bases of sustainable development; (II) Environmental pollution: energy and environment, the aquatic environment, the terrestrial environment, the atmospheric environment; (III) Sustainable development: basic concepts, economy and environment, legal and institutional aspects, assessment of environmental impacts, environmental management (Braga and Hespanhol 2005).

6    According to the General Directory of CNPq (National Council for Scientific and Technological Development) research groups, in the year 2023, there are 2349 GPs that will be registered in Brazil related to ecology, in different Educational Institutions and areas of study (https://dgp.cnpq.br/dgp/faces/consulta/consulta_parametrizada.jsf, accessed on 28 October 2023).

7    An initiative of the Research Group "transdisciplinarity, integral ecology and Socio-environmental Justice", the three-volume work brings together authors from various areas of knowledge and Higher Education institutions, including theology.

8    In Brazil, traditional communities of Afro-descendants who live in community territories are called "quilombolas". Quilombos originated from groups of slaves who fled farms and established a way of living that was cooperative and linked to nature, maintaining their traditions and values. According to official data from the Palmares Foundation, in 2021, there were at least 3450 recognized quilombola communities in the country, with community ownership of the territory. There are several similar communities of Afro-descendants in other countries in Latin America and the Caribbean, such as Colombia, Ecuador, Suriname, Honduras, Belize and Nicaragua.

9    One can add, in the Protestant sphere, the positions of the World Council of Churches, especially from the assembly in Seoul and Vancouver, and the adoption of the Justice, Peace and Integrity of Creation Program (JPIC), year 1990.

10    It is necessary to distinguish between epistemic and axiological anthropocentrism. As human beings, we see and understand the world and other creatures based on our species. We interpret the meaning of other beings with the hermeneutic keys that are our own, based on our sensory mechanisms. Axiological anthropocentrism concerns the values we cultivate, in our relationship with biotic and abiotic beings, as well as with the planet's community of life.

11    F. CAPRA also makes a social expansion of the "paradigm" category, defining it as "a constellation of conceptions, values, perceptions and practices shared by a community, which gives shape to a particular vision of reality, which constitutes the basis of the way the community organizes itself" (https://www.docsity.com/pt/ecologia-profunda-um-novo-paradigma/4728071/, accessed on 28 October 2023).

12    We adopt a slightly different position from E. Conradie. We consider these steps of Ecotheology to be an ongoing process. Therefore, we do not place them as an "agenda" or "tasks", but rather as open paths that are taken.

13    These are available free of charge on the Amerindia network website, especially in the "Publications", "blogs" and "Integral Ecology" sections.

14    https://repam.org.br/wp-content/uploads/2022/05/Revista-Ecotelogia-No-1-PDF16MAIO22.pdf (accessed on 28 October 2023).

15    https://cimi.org.br/ (accessed on 29 October 2023).

16    https://iglesiasymineria.org/ (accessed on 29 October 2023).

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
