# Peer review of "Latin American and Caribbean Ecotheology: A Kaleidoscope"

_religions, doi:10.3390/rel14121500_

Round 1

Reviewer 1 Report

Comments and Suggestions for Authors

Congrats on a fine analysis of the theoretical and theological basis of an emerging ecotheology and its relationship with Liberation theology. I read this paper cover to cover immediately. The clarity of language, the progression of the argument, and the evidence offered present one of the finer papers that I have reviewed and read in these past years. Hat tip!

I offer two critiques, one grammatical and one substantive challenge in the argument.  Let's start with the grammatical: please eliminate and rewrite any beginning of the sentence with "This is" to one that identifies the subject of the demonstrative pronoun, "this."  You shift the burden of interpretation to the reader when you do not identify the noun referred when writing "This means...." Simply identify the noun. 

Next, your argument intends to offer a call to "decolonize" Western thinking as if we can move into an idyllic environment void of such a heritage...see "getting rid" (line 465 thru 470) and "let go of corrupted practices" (line 513).  Your argument later acknowledges and limits this prior assertion by stating that we cannot "take refuge in an idyllic environment...(533)."    The next sentence left me confused and I don't capture the intent of the active verb, "triggering" (533-34).  Do you mean "unmask" the illusory myth and make transparent this misstep in our colonized mentality?  The following paragraphs on the relationship between the particular and universal accomplish your intent to argue that we dialogue with the Other given the particularities of our moment in time with all universal claims. That dialogue recognizes implicitly the fallenness of both subjects, particularly those from the West who in times past imposed universal claims and did not listen to competing claims from the Other.  The phrases "getting rid" and "letting go" do strike me as call to retreat to some pristine encounter that denies the particular moment of each space-history.  Your subsequent writing clarifies, even contradicts these prior assertions.  

All is all, these critiques are minimal compared to the joy of reading your insights and conclusions. Many thanks.

Comments on the Quality of English Language

Excellent except for a grammatical "tick" mentioned above that the author can easily remedy.

Author Response

Thank you very much for the detailed observations and nuances. These revisions for our text provided us with the opportunity to clarify our argument. We appreciate the great attentiveness to detail provided by your review that without a doubt allowed for greater coherence in our text. 

Reviewer 2 Report

Comments and Suggestions for Authors

The article is an intersting text on ecoheology presented from the perspective of Latin Amercian and carribean point of view. The Author presents an actuall question in a clearly way. The article is well structured and about each parts talks in an exhaustive way, so the reader can receive an interesting text on kailedoscope of ecotheology. The article can be publish in the present form.

Author Response

Thank you for your observations. 

Reviewer 3 Report

Comments and Suggestions for Authors

The article is very good. I could not add anything to it, nor do I have corrections to do about the contents.

Since I am not an English native speaker, I am not able to say whether English is perfect. It seems to be very good, anyway.

I noted on line 886 Thomaz Kuhn (if I am not mistaken, it should be Thomas Kuhn), on line 345 res extenso (it should be res extensa). I do not know whether the question mark of line 421 is intentional.

Author Response

Thank you for the observations. We have made the required changes for lines 886, 345 and 421.